# Design of a Tri-Band Wearable Antenna for Millimeter-Wave 5G Applications

**DOI:** 10.3390/s22208012

**Published:** 2022-10-20

**Authors:** Sarosh Ahmad, Hichem Boubakar, Salman Naseer, Mohammad Ehsanul Alim, Yawar Ali Sheikh, Adnan Ghaffar, Ahmed Jamal Abdullah Al-Gburi, Naser Ojaroudi Parchin

**Affiliations:** 1Department of Signal Theory and Communications, Universidad Carlos III de Madrid (UC3M), Leganes, 28911 Madrid, Spain; 2Department of Electrical Engineering and Technology, Government College University Faisalabad (GCUF), Faisalabad 38000, Pakistan; 3Department of Information Processing and Telecommunications Laboratory (LTIT), Faculty of Technology, University of Bechar, Bechar 08000, Algeria; 4Department of Information Technology, University of the Punjab Gujranwala Campus, Gujranwala 52250, Pakistan; 5Department of Electrical and Computer Engineering, University of Delaware, Newark, DE 19716, USA; 6Department of Electrical and Electronic Engineering, Auckland University of Technology, Auckland 1010, New Zealand; 7Department of Electronics and Computer Engineering (FKEKK), Center for Telecommunication Research and Innovation (CeTRI), Universiti Teknikal Malaysia Melaka (UTeM), Durian Tungal 76100, Malaysia; 8School of Engineering and the Built Environment, Edinburgh Napier University, Edinburgh EH10 5DT, UK

**Keywords:** tri-band antenna, mm-wave antenna, 5G antenna, wearable antenna

## Abstract

A printed monopole antenna for millimeter-wave applications in the 5G frequency region is described in this research. As a result, the proposed antenna resonates in three frequency bands that are designated for 5G communication systems, including 28 GHz, 38 GHz, and 60 GHz (V band). For the sake of compactness, the coplanar waveguide (CPW) method is used. The overall size of the proposed tri-band antenna is 4 mm × 3 mm × 0.25 mm. Using a watch strap and human tissue, such as skin, the proposed antenna gives steady results. At 28 GHz, 38 GHz, and 60 GHz, the antenna’s gain is found to be 5.29 dB, 7.47 dB, and 9 dB, respectively. The overall simulated radiation efficiency is found to be 85% over the watch strap. Wearable devices are a great fit for the proposed tri-band antenna. The antenna prototype was built and tested in order to verify its performance. It can be observed that the simulated and measured results are in close contact. According to our comparative research, the proposed antenna is a good choice for smart 5G devices because of its small size and simple structure, as well as its high gain and radiation efficiency.

## 1. Introduction

The increase in the number of users will cause a need for new communication systems with high data rates, low costs, efficiency, and simplicity [1]. A viable choice for these systems is the fifth generation (5G) of communication technology [2,3]. The 28 GHz and 38 GHz millimeter-wave (mm-wave) frequency bands are promising for 5G wireless communication because of the low absorption rate and minimal error for both line of sight (LOS) and non-line of sight (NLOS) systems [4,5,6]. For satellite communication systems, the 60 GHz V-band is a suitable frequency range along with Wi-Fi. At 5G mm-wave frequencies, an antenna’s geometrical structure and small size play a critical role in the communication system. Antennas for 5G must also have high gain and efficiency due to increased air absorption and attenuations [7,8,9]. However, the 5G antenna’s strong gain is compromised due to its compact size. Consequently, an antenna with strong radiation properties is required for the 28 GHz, 38 GHz, and 60 GHz millimeter-wave frequency bands.

Much research was carried out recently on mm-wave frequency for single, dual, or multiband operation [10,11,12,13,14,15,16,17,18,19]. The wire-bond antenna for V-band applications was presented in [10] with a circularly polarized capability. It was small and capable of transmitting from 51 GHz to the upper end of the frequency range. However, the proposed antenna has a complex geometrical structure and low gain (0.8 dBi). Another V-band mm-wave antenna was proposed to achieve a high gain of 14 dBi and a broad bandwidth of 57 GHz to 66 GHz [11]. Though the proposed antenna has high gain although it is large in size. A mm-wave microstrip fed antenna with a single resonant band between 34.1 and 38.9 GHz is described in [12]. Directivity values ranged from 6 dBi to 8 dBi for single antenna setup and parasitic patches, respectively, in the given structure. A tri-band antenna for 5G mm-wave applications was proposed in another paper [13]. On the first layer, microstrip feed is provided, and on the second, proximity-linked parasitic patches are used. At a maximal gain of 5.66 dBi, this antenna resonantly resonates at 45.3, 55.7, and 66 GHz. Similarly, [14] presents a mm-wave communication system triple band antenna with a very basic configuration. With very limited impedance bandwidth, this antenna achieved peak gains of 6.5, 7, and 5 dBi, respectively, at 24 GHz, 28 GHz, and 38 GHz. As another example of an inset feed, a three-band slotted millimeter-wave antenna was published [15]. The antenna has a maximum gain of 6.3 dBi across the resonant bands. Other research [16] proposed a dual-band mm-wave antenna for 5G communication systems operating at 28 GHz and 38 GHz. In terms of peak gain, the antenna covers the 26.65–29.2 GHz and 36.95–39.05 GHz bands, respectively, with a peak gain of 1.27 dBi and 1.83 dBi. A high-gain dual-band antenna described in [17] may operate at 28 and 38 gigahertz. The antenna, on the other hand, has a huge substrate. Ref. [18] presents an antenna design for 28/38 GHz 5G mm-wave frequencies with a T-shaped dual band antenna and a triangular-shaped slot on the bottom layer for antenna performance optimization. This highest gain of 5.75 and 7.23 dBi, respectively, is seen at 28 and 38 gigahertz [19]. A new microwave and mm-wave multiband triangular patch antenna is described. In order to cover the 23–28 GHz band, this antenna has an overall gain of 5.85 dB, although its total size is rather huge. At mm-wave frequencies, researchers have an enormous difficulty in constructing low-profile [20] and small antennas that have broad bandwidth, high gain, and multiband properties.

A low-profile, tri-band, high gain monopole antenna for 28 GHz, 38 GHz, and 60 GHz mm-wave frequency bands is discussed in this research, considering the previously described difficulties and constraints. Monopole antennas [21] were chosen for this project because of their many advantages, including their small size, low profile, and simple geometry. As a result of its excellent radiation and tri-band properties, the proposed antenna is well-suited to mm-wave communication systems. According to the following structure: in Section 2, the design methodology, phases for the proposed multiband antenna, as well as a parametric analysis, are shown. In Section 3, the effective parameters of the antenna are looked at. In Section 4, the results of this study are compared to the results of other studies in the same field. Section 5 ends the paper with a list of the sources that were used.

## 2. Antenna Design Methodology

### 2.1. Antenna Design Stages

Figure 1 depicts the four steps to design the final optimal antenna shape. First, a rectangular monopole antenna (ANT I) with CPW feeds is developed, which resonates at a 30 GHz resonant frequency, as shown in Figure 1b. In design step II (ANT-II), the proposed rectangular monopole antenna is truncated to resonate at 28 GHz, with an impedance bandwidth of 26–31 GHz. To generate the additional 60 GHz frequency band, a U-shaped slot is introduced in the ANT II, as illustrated in Figure 1a. The addition of the U-shape in the proposed antenna design yielded into two frequency bands, in which one band is enough to span the needed frequency range for 5G communication systems. In the final step (ANT IV), a C-shaped slot is created in the upper edge of the radiator to resonate the proposed antenna at a 38 GHz frequency band. The stub loading approach is not only used for enhancing the impedance bandwidth, but also improves the impedance matching at the target frequency [22,23,24]. The resonating frequencies are then optimized in certain band spectrums. Figure 1b shows the return loss of the proposed antenna during the design steps, and resonates at 28, 38, and 60 GHz frequency bands.

Here is a breakdown of the design process for the triband mm-wave antenna:

The basic antenna design consists of a 50-ohm CPW feedline and a printed rectangular monopole (ANT I in Figure 2a). The effective dielectric constant, by neglecting the thickness of the conductor [17], is given in (1). This CPW configuration, seen in Figure 1, has ground on both sides of the feed strip and no back conductor. Using Equations (1)–(7), the essential parameters of the antenna with CPW ground plane can easily be calculated.
(1)$$Z_{o}=\frac{30\pi K^{\prime }\left(k\right)}{\sqrt{\varepsilon_{re}}\;\cdot \;K\left(k\right)}$$
(2)$$K^{\prime }\left(k\right)=K\left(k^{\prime }\right)$$
(3)$$k^{\prime }=\sqrt{1-k^{2}}$$
(4)$$k=\frac{wf}{wf+2\left(g\right)}$$
(5)$$\varepsilon_{re}=\frac{\varepsilon_{r}+1}{2}\;\left[P+Q\right]$$
here,
(6)$$P=\tan h\left\{1.785\log\left(\frac{hs}{g}\right)+1.75\right\}$$
(7)$$Q=\frac{K\left(g\right)}{hs}0.04-0.7k+0.01\left(1-0.1\varepsilon_{r}\right)\left(0.25+k\right)$$
where *K(k*) denotes the first sort of elliptical integral and *K(k’*) its complement. The feed strip width (*wf*) is 0.26 mm, and the gap (*g*) between the feed strip and the ground is 0.03 mm, with *hs* being the substrate thickness.

### 2.2. Parameteric Analysis

A parametric analysis of the important parameters is carried out to get the optimum results with a broad bandwidth and to enhance impedance matching. Tri-band operation may be achieved at the ideal height of *H =* 0.25 mm, with 28 GHz, 38 GHz, and 60 GHz all supported. It is critical to consider the slot’s length and width while designing an antenna that will provide the desired resonance frequencies, broad impedance, and high impedance matching. For the truncated edges of the slot’s width (*W_6_*), parametric analysis is shown in Figure 2. Impedance mismatches are seen in the 28 GHz range when W_6_ is raised to 0.68 mm; the other two bands remain the same. The bandwidth and impedance mismatch are both reduced when *W_6_* is set to 0 mm, however. Because of this, it is proposed that the best value for *W_6_* in terms of bandwidth and triband functioning at mm-wave frequencies is 0.68 mm. The parametric analysis of the upper C-shaped slot is varied from 0.8 mm to 1.2, as shown in Figure 2b. It can be seen that by varying the values of the C-shaped slot, the lowest band (28 GHz) and the upper band (60 GHz) remained unchanged, while the middle band (38 GHz) changed. Similarly, the V-band frequency ranges from 56 to 61 GHz when the U-shaped slot’s width is adjusted to 1 mm. As indicated in Figure 2c, the other two bands show signs of mismatching and shifting to undesirable frequencies. Decreased impedance matching at resonance bands and a wider V-band are both found when W_7_ is reduced to 1.4 mm. This has a substantial impact on antenna performance. Again, impedance mismatching, and band narrowing occur when W_7_ is further lowered to 1 mm. In other words, the optimal U-slot width displays the optimum performance in terms of bandwidth and reflection coefficient when it is 1 mm long. The parametric study for the length and width of the ground planes ‘*Lg*’ and the ‘*wg*’ is shown in Figure 2d,e. With the help of parametric analysis, it is analyzed that each slot in the radiator is independent to the other, which means that by varying the length and width of one slot, we can easily shift one frequency by setting the rest of the frequencies.

### 2.3. Antenna Geometry

The proposed antenna’s top, back, and side views are shown in Figure 3a–c. The antenna design is modelled using Rogers R/Duroid5880 with a relative permittivity of 2.2, and loss tangent of 0.12. The thickness of Rogers material is 0.25 mm. This material has a low dielectric constant and low dielectric loss, which makes it suitable for high frequency/broadband applications. This material is isotropic and absorbs low moisture. The overall size of the proposed antenna is L_s_ × W_s_ × H_s_. Copper is used on the top side of the substrate and has an effective length of λ/2. A mm-wave tri band antenna is proposed which resonates at 28 GHz, 38 GHz, and 60 GHz operating frequencies. The U-shaped and C-shaped slots are inserted in the radiator, and the center side of the rectangular monopole antenna is truncated. The truncated side, top invert C-shaped slot, and bottom U-shaped slots are used to resonate at 28 GHz, 38 GHz, and 60 GHz, respectively. A Computer Simulation Technology (CST) Studio suite is used to model the antenna design. Several antenna parameters are optimized during the design process to provide the best possible radiation characteristics. Table 1 indicates the proposed dimension parameters of the optimized antenna.

### 2.4. Surface Current Distribution

Figure 4 shows surface current distribution of the proposed antenna at various operating frequencies. Resonance frequencies (28 GHz and 38 GHz) are produced at the bottom side of the proposed antenna because of its longer effective electrical length, as shown in Figure 4a,b, respectively. As illustrated in Figure 4, the truncated side of the radiator has a major contribution in producing the 28 GHz frequency band. As shown in Figure 4b, the upper C-shaped slot helps to generate the 38 GHz frequency band, and the bottom U-shaped slot helps to operate at the 60 GHz frequency band.

### 2.5. Bending Analysis of the Proposed Antenna

In order to verify the antenna’s *S_11_*, it is required to have bending analysis along the horizontal and vertical axes of the antenna, which is made up of a flexible Roger RT5880 substrate. The bending of the antenna on wearable situations is critical. Antenna bend analysis in free space along the *x-* and y-axis is described in this section. Using a cylindrical form with a diameter of 30 mm, 60 mm, and 90 mm, the bending analysis of the antenna is checked with simulations.

#### 2.5.1. Bending along *X*-Axis

When it comes to studying *S_11_* behavior when the antenna is in bent condition, the antenna’s bending radius along the *x*-axis (Bx = 30 mm to 90 mm) is chosen. As shown in Figure 5, when the antenna is bent along the *x*-axis, the simulated *S_11_* results are compared over different radii. It is realized that the antenna has stable *S_11_*, even if it bent at the radius of 30 mm, 60 mm, and 90 mm.

#### 2.5.2. Bending along *Y*-Axis

While it comes to researching the behavior of the *S_11_* parameter when the antenna is in a bent state, the bending radius along the *y-*axis of the antenna is selected to be between 30 mm and 90 mm. When the antenna is bent along the *y-*axis, the results of the simulated *S_11_* are compared across various radii, as illustrated in Figure 6. It was determined that the antenna maintains a steady *S_11_* even when bent at radii of 30 mm, 60 mm, and 90 mm, respectively.

## 3. Experimental Results and Discussions

The measured and simulated results are presented in this section to validate the performance of the proposed antenna design. The antenna prototype is fabricated on ROGERS 5880 substrate, as shown in Figure 7a. A 50 ohm-thin block end launch connector (withwave^®^) is used to measure the return loss and radiation pattern. The maximum working frequency of this connector is 67.5 GHz [25]. The comparison between the simulated and measured return loss in shown in Figure 7b. As shown in the Figure 7b, the proposed antenna resonates at three operating modes, with operating frequencies of 28, 38, and 60 GHz, respectively. The impedance bandwidth for 28, 38, and 60 GHz are 21 GHz to 31 GHz, 37 to 39 GHz, and 56 to 61.5 GHz, respectively. There is a decent agreement between simulation and measurement results, however, the little variance is due to inaccuracies in the manufacturing and measurement equipment.

### 3.1. Measurement Setup

The R&S ZVA110 Vector Network Analyzer (VNA) (Rohde & Schwarz, Munich, Germany) is utilized to investigate and verify the proposed tri-band antenna’s frequency domain reflection coefficient. Far-field radiation parameters of the proposed design are analyzed using a shielded RF anechoic chamber. A standard 24 dBi horn antenna (SGH-series) is utilized as a transmitter while the proposed antenna design is used as a receiver. Transmission losses and open air may affect the signal produced by a voltage-controlled oscillator (VCO). Measurement set-up employed high-gain power amplifiers, which can boost the produced signal and efficiently transmit the radio signal from transmitter to receiver.

### 3.2. Radiation Pattern

Figure 8a–c illustrates the radiation pattern of the proposed antenna at various resonance frequencies. The omnidirectional pattern in the H-plane and the elliptical pattern in the E-plane can be seen in both the 28 GHz and 38 GHz radiation patterns. For frequencies between 28 GHz and 38 GHz, a high correlation is observed between simulated and measured results. An omnidirectional pattern along the H-plane and a broadsided pattern along the E-plane is observed for the 60 GHz frequency because of the 40 GHz measurement constraint.

### 3.3. Gain and Radiation Efficiency

Figure 9a shows the simulated and measured gain and efficiency of the proposed design operating at a variety of frequency bands, including 28 GHz, 38 GHz, and 60 GHz. The measured gain is 1.8 dB and 1.6 dB at 28 GHz and 38 GHz, respectively. The simulated gain of the antenna in free space is 2.1 dB, 1.9 dB, and 5.1 dB at 28, 38, and 60 GHz, correspondingly. Most of the input power is lost as radiation by an antenna with a high radiation efficiency. Most of the energy is lost as internal losses in a low-efficiency antenna or is reflected away owing to an impedance mismatch.

For antenna efficiency, also called radiation efficiency, the efficiency of an antenna is defined as the ratio of the input power to the output power (8):(8)εR=PradPin

The simulated and measurement results show that the antenna’s efficiency across all frequency bands is lower than 90%, as shown in Figure 9a. Figure 9b indicates the front-to-back ratio (FBR) of the proposed antenna in free space. The value of the FBR is found to be more than 15 dB at 28 GHz and 38 GHz, while more than 28 dB for 60 GHz.

## 4. Simulation Results of Tri-Band Antenna on Human Body

The proposed antenna is tested on a watch strap made of a rubber material and on human tissue, such as skin, to see human closeness effects. The small size antenna is maintained near the skin (ɛ_r_ = 41.4 and σ = 0.88 s/m) [20]. The size of the phantom box is equal to 20 × 20 mm^2^. Figure 10a shows the proposed structure when the antenna is in proximity to the human tissue, and the simulated return loss is shown in Figure 10b.

Figure 11a–c describes the radiation pattern of the proposed tri-band millimeter-wave antenna at 28 GHz, 38 GHz, and 60 GHz frequency bands. The elliptical radiation pattern in the E and H-plane is observed at 28 GHz, as shown in Figure 11a, while the broadsided directional pattern can be seen along the E and H-plane at 38 GHz. The proposed antenna shows the directional radiation pattern at 60 GHz, as shown in Figure 11c. Figure 12 illustrates the simulated gain and efficiency of the proposed antenna operating at a variety of frequency bands, including 28 GHz, 38 GHz, and 60 GHz. The simulated gain is 5.3 dB, 7.5 dB, and 9 dB at 28 GHz, 38 GHz, and at 60 GHz frequency bands, respectively. The antenna’s efficiency across all frequency bands is lower than 80%.

### Specific Absorption Rate (SAR)

Electromagnetic radiation may pose health concerns to humans, and these risks are quantified using the SAR. The following equation describes the link between input power and SAR (9):(9)$$\mathrm{SAR}=\frac{\sigma \left|E^{2}\right|}{\rho }$$
where ‘σ’ and ‘ρ’ are the thermal conductivity (S/m), and mass density (kg/m^3^), respectively while electric field intensity (V/m) are denoted by E. According to the equation below (10), the electric power intensity varies proportional to the signal power.
(10)$$\mathrm{Power}\;(W/m^{2})=\frac{{\left(E\left(V/m\right)\right)}^{2}}{377}$$

It is possible to conduct SAR simulations if the antenna is retained over the phantom. An average of 0.5 W of power is used to determine a device’s SAR based on the IEEE/IEC 6270-1 standard. Figure 13 shows the SAR values of 0.9 W/kg at 28 GHz, 0.62 W/kg at 38 GHz, and 0.74 W/kg at 60 GHz over 1 g of mass tissue. While over 10 g of mass tissue, SAR values are 1.49 W/kg, 1.17 W/kg, and 1.47 W/kg at 28 GHz, 38 GHz, and 60 GHz, respectively.

## 5. State-of-the-Art Works Comparison

For the modelling of the proposed antenna, CST MW Studio is used for designing and simulating the proposed antenna. Figure 11 illustrates the two-dimensional radiation pattern produced by the antenna integrated within the smart watch. Antenna stability can be shown in Figure 5, Figure 6, and Figure 10 when it is simulated in free space and on the watch strap. A total radiation efficiency of over 82% was found for the antenna and may be tested both in free space and on the rubber strap. Table 2 compares the proposed antenna to literary studies that were recently published. This piece has an antenna that is smaller than most of the others, making it stand out from the others. As a result, the antennas described in [13,14,15] have a limited bandwidth and poor strength, even if they are tri-band compact. Researchers [16,17,18] reported huge antennas with dual-band operation and a limited bandwidth. Thus, the triband antenna’s superior radiation properties, such as a broad bandwidth, large antenna gain, greater radiation efficiency, as well as a simple and a compact size, make it suitable for 5G mm-wave devices. Figure 14 describes the comparison of the *S*_11_ between the free space and on the phantom.

## 6. Conclusions

This paper presents a very compact size tri-band antenna operating at mm-wave frequency bands of 28, 38, and 60 GHz. A CPW-fed radiating monopole is printed on a flexible Roger 5880 substrate. The dimensions of the design are as follows: 4 × 3 × 0.25 mm^3^. The proposed design is simulated in free space and on the rubber material with a skin tissue for the use of a smart watch. It is observed that over different layers of phantom, the total radiation efficiency is found to be more than 85%, and the gain of the proposed antenna at 28 GHz, 38 GHz, and 60 GHz is 5.29 dB, 7.47 dB, and 9 dB, respectively. SAR values are found to be 1.49 W/kg, 1.17 W/kg, and 1.47 W/kg at the frequencies of 28 GHz, 38 GHz, and 60 GHz. Furthermore, a comparison of the proposed antenna to current work shows that the proposed design is a perfect candidate for future communication systems.

## Figures and Tables

**Figure 1 sensors-22-08012-f001:**
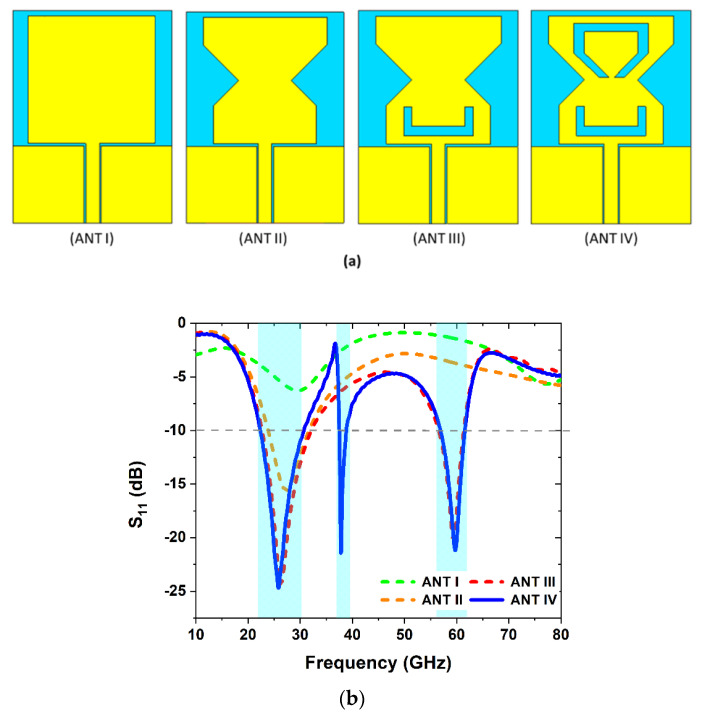
(**a**) Proposed design steps of the antenna. (**b**) Antenna return loss comparisons at various phases of design.

**Figure 2 sensors-22-08012-f002:**
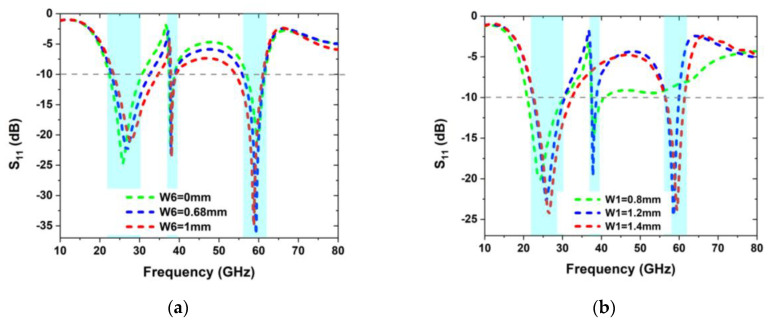
Parametric Analysis of the optimized parameters; (**a**) variation in *W_6_*, (**b**) variation in *W_7_*, (**c**) variation in *W_1_*, (**d**) variation in *L_g_*, and (**e**) variation in *W_g_*.

**Figure 3 sensors-22-08012-f003:**
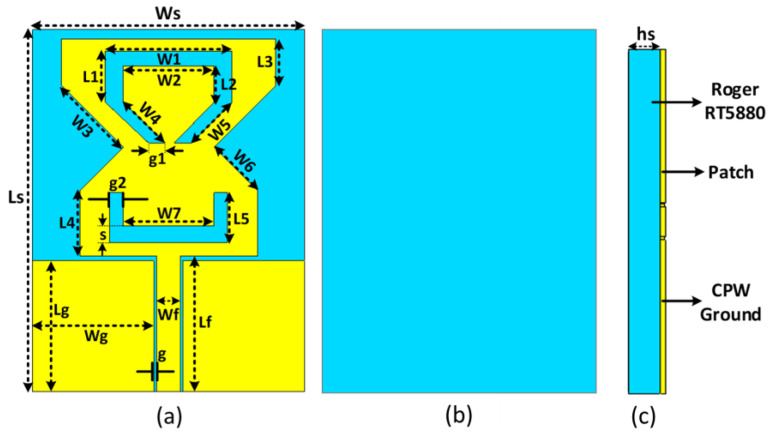
Microstrip feedline and an umbrella-shaped radiator comprise the antenna’s geometry; (**a**) front view, (**b**) back view, and (**c**) side view.

**Figure 4 sensors-22-08012-f004:**
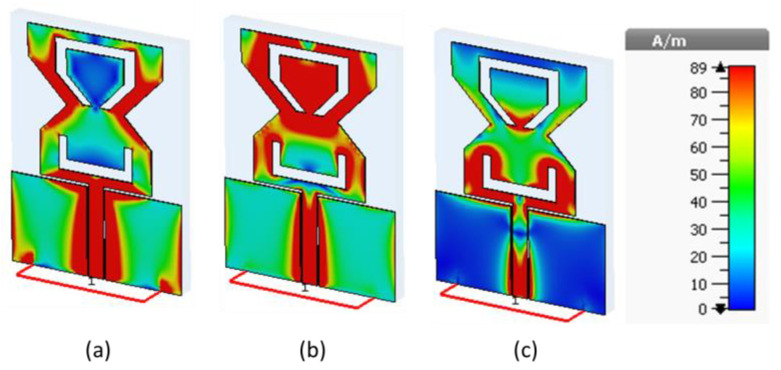
Surface current distribution at specified frequencies; (**a**) 28 GHz, (**b**) 38 GHz, and (**c**) 60 GHz.

**Figure 5 sensors-22-08012-f005:**
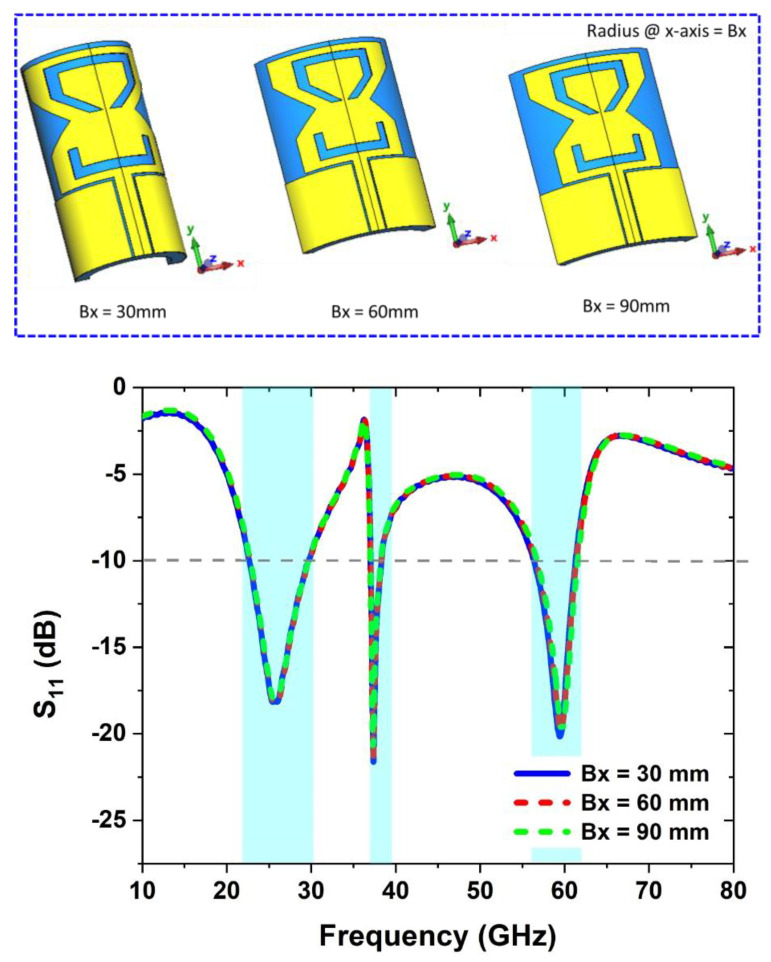
Antenna bending analysis along *x*-axis.

**Figure 6 sensors-22-08012-f006:**
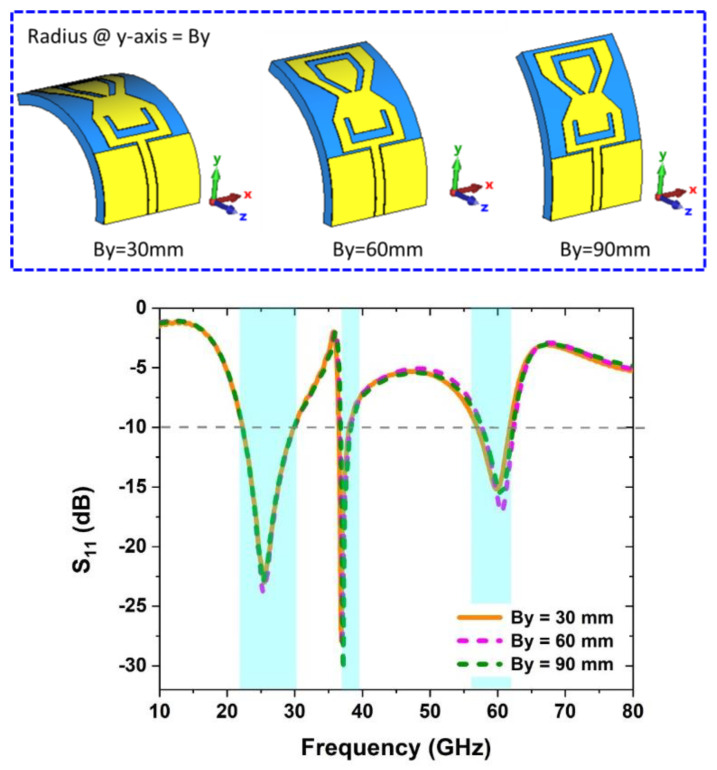
Antenna bending analysis along *y*-axis.

**Figure 7 sensors-22-08012-f007:**
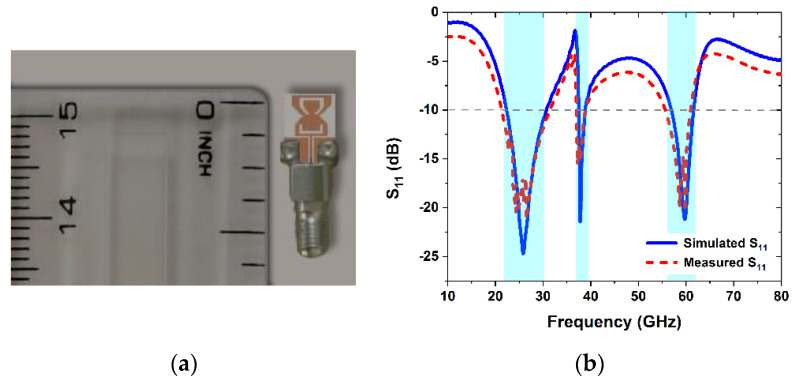
(**a**) The prototype of the proposed antenna; (**b**) comparison between the simulated and measured return loss.

**Figure 8 sensors-22-08012-f008:**
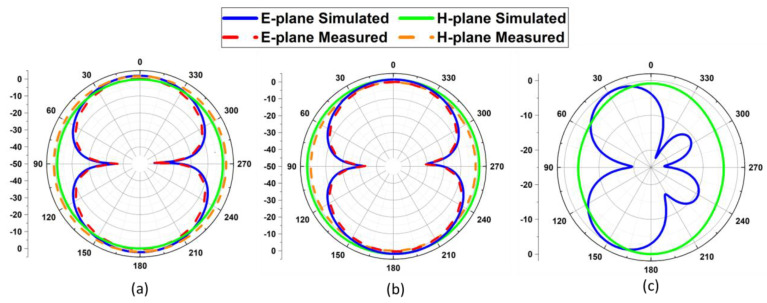
The 2D pattern of the proposed design; (**a**) at 28 GHz, (**b**) at 38 GHz, and (**c**) at 60 GHz.

**Figure 9 sensors-22-08012-f009:**
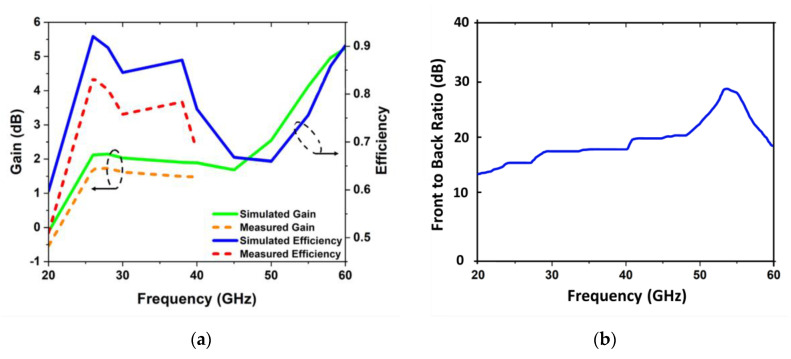
(**a**) Efficiency and gain curves of radiation in relation to different operational frequency bands in free space, (**b**) front-to-back ratio (FBR).

**Figure 10 sensors-22-08012-f010:**
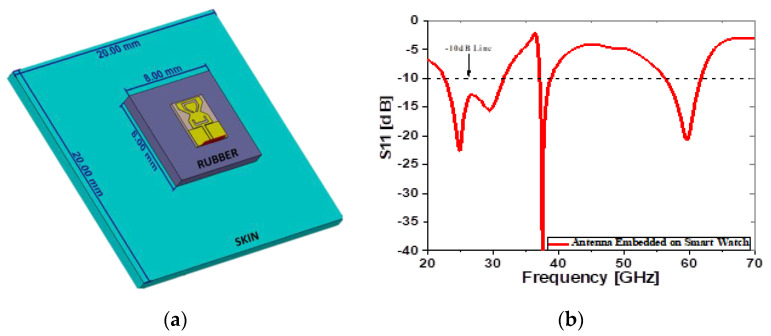
(**a**) Proposed antenna over rubber and skin tissues; (**b**) simulated return loss.

**Figure 11 sensors-22-08012-f011:**
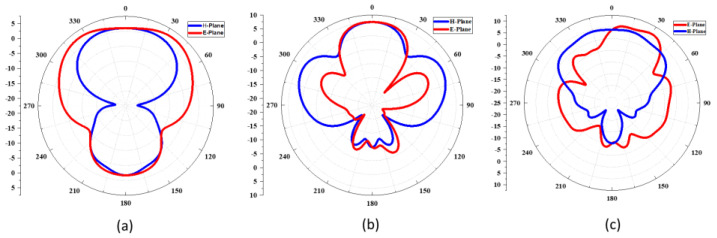
Antenna’s 2D radiation pattern over phantoms at the following frequencies: (**a**) 28 GHz, (**b**) 38 GHz, and (**c**) 60 GHz.

**Figure 12 sensors-22-08012-f012:**
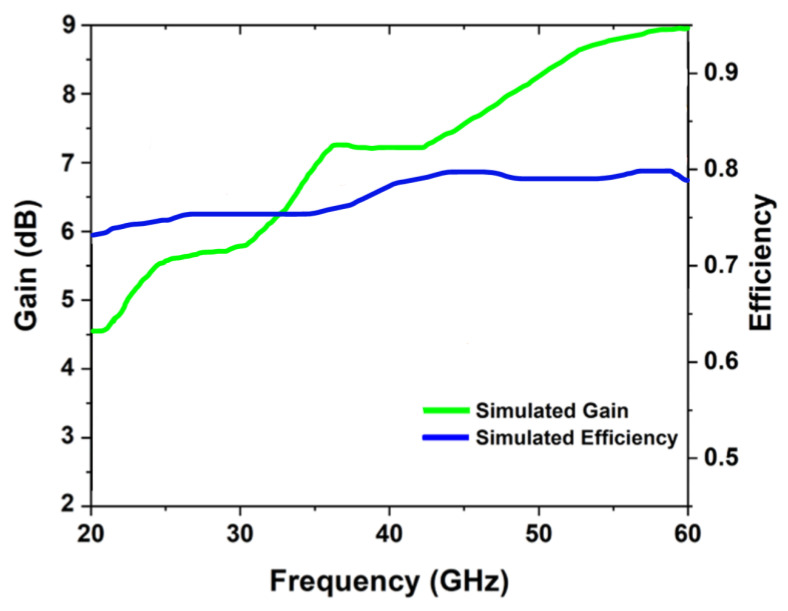
Efficiency and gain curves at operational frequency bands over phantoms.

**Figure 13 sensors-22-08012-f013:**
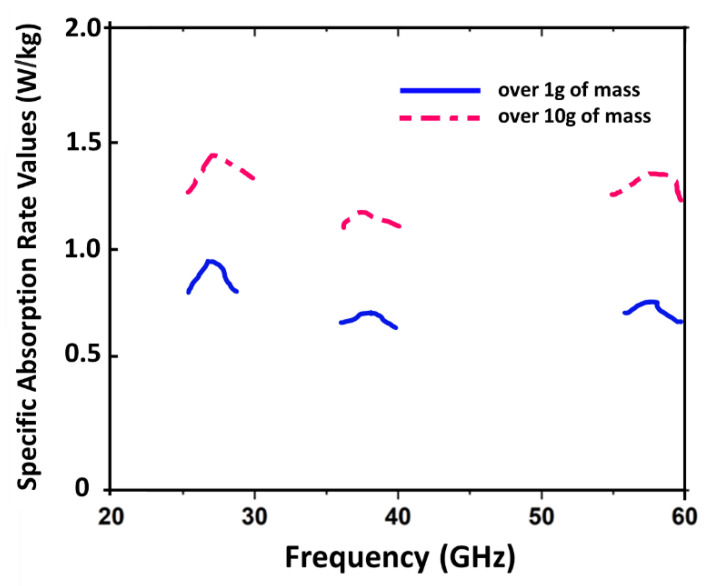
SAR values vs. frequency graph over phantom using mathematical equations.

**Figure 14 sensors-22-08012-f014:**
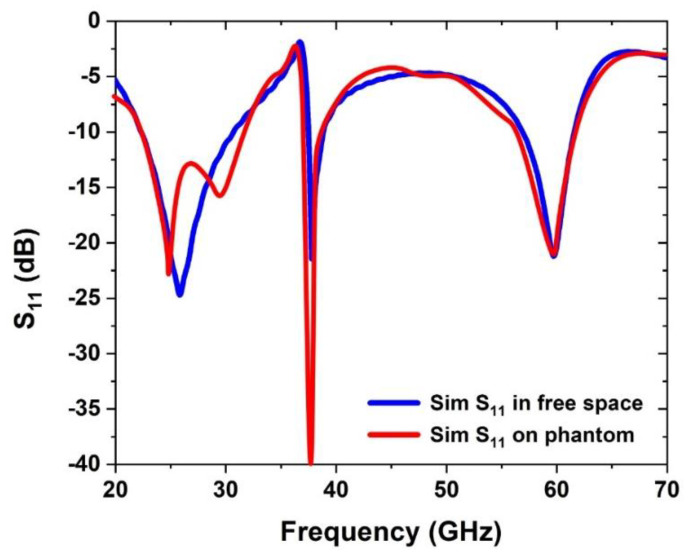
Comparison of the *S_11_* in free space and on phantom.

**Table 1 sensors-22-08012-t001:** Optimized dimensions of the proposed design.

Parameters	Values (mm)	Parameters	Values (mm)
*Ls*	*4.0*	*Ws*	*3.0*
*Lg*	*1.45*	*wg*	*1.34*
*Lf*	*1.5*	*wf*	*0.26*
*L1*	*0.56*	*w1*	*1.4*
*L2*	*0.4*	*w2*	*1.0*
*L3*	*0.52*	*w3*	*0.96*
*L4*	*0.72*	*w4*	*0.65*
*L5*	*0.55*	*w5*	*0.64*
*g1*	*0.19*	*w6*	*0.68*
*g2*	*0.15*	*w7*	*1.0*
*g*	*0.03*	*hs*	*0.25*
*s*		*0.18*	

**Table 2 sensors-22-08012-t002:** Analysis of the planned study in comparison to relevant work.

Ref. No.	Antenna Size (mm^3^)	Bandwidth (GHz)	No. of Bands	Operating Frequency (GHz)	Peak Gain (dB)
[13]	10 × 5 × 0.51	3, 5, 3	Tri-band	45, 57, 66	5.6
[14]	4 × 5 × 0.2	0.5, 0.9, 0.4	Tri-band	24.4, 28, 38	6.5, 7, 5
[15]	8 × 8 × 1.6	2.5, 1.6, 7	Tri-band	26, 35, 54	5.8, 4.5, 6
[16]	14 × 12 × 0.38	2.6, 2.1	Dual band	28, 38	1.27, 1.83
[17]	55 × 110 × 0.05	0.8, 1.5	Dual band	28, 38	7.9, 8.2
[18]	20 × 20 × 1.95	4.8, 3.6	Dual band	28, 38	5.7, 7.2
[19]	8 × 8 × 0.79	2.3, 5.1, 15	Tri-band	28, 38, 55	6.6, 7, 7.35
**[This work]**	**4 × 3 × 0.25**	**10, 1.5, 6**	**Tri-band**	**28, 38, 60**	**5.29, 7.49, 9**

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
