# Peer review of "Design of a Tri-Band Wearable Antenna for Millimeter-Wave 5G Applications"

_sensors, 2022, doi:10.3390/s22208012_

Round 1

Reviewer 1 Report

The paper presents 28 GHz, 38 GHz, and 60 GHz antenna thanks to monopolies and slots modes. The idea is not original, but the implementation in the mm-wave range gives some merit.

The real problems with the proposed structure are:

1- the incoherence of the propagation at the 3 bands. As shown in fig. 6 at the angle 0 the 60 GHz antenna has a lower gain. (This problem is, unfortunately, common in the multi-band antenna area).

2-The presented antenna is not patch: the field distribution in fig 4, shows clearly that the first frequency is edge mode, manual mode for the 2d and slot mode for the third frequency. (The higher mode -2ed which should not radiate) and probably this antenna radiated at 2 or more bands lower than 24 GHz. The structure was designed by chance as clearly demonstrated by the bandwidth of each band. 

Moreover, with the incoherence of the beam direction, this structure shows also incoherence of the polarization.
The H plan and E plan are considered here based on patch mode, which is not the case.

3- The measured prototype: it is clear that the addition of the connector as shown in fig 5 will affect the pattern, the efficiency as well as the S11.

 This type of connector will add a minimum of 1.5 dB loss per the second band and 3 dB at 60 GHz.This loss plus the radiation how the measured efficiency is 85 %.

5- To be considered multi-band, independent frequency control would have to be clearly demonstrated.

Author Response

Reviewer 1

The paper presents 28 GHz, 38 GHz, and 60 GHz antenna thanks to monopolies and slots modes. The idea is not original, but the implementation in the mm-wave range gives some merit. The real problems with the proposed structure are:

Comment_1: The incoherence of the propagation at the 3 bands. As shown in fig. 6 at the angle 0 the 60 GHz antenna has a lower gain. (This problem is, unfortunately, common in the multi-band antenna area).
Response_1:
We are thankful to the reviewer for pointing out the main concern. The simulated gain of the antenna at 60 GHz is found to be almost 5.1 dB. But in the future work we will try to extend the work in MIMO or in antenna array to increase the gain of the antenna.

Comment_2: The presented antenna is not patch: the field distribution in fig 4, shows clearly that the first frequency is edge mode, manual mode for the 2d and slot mode for the third frequency. (The higher mode -2ed which should not radiate) and probably this antenna radiated at 2 or more bands lower than 24 GHz. The structure was designed by chance as clearly demonstrated by the bandwidth of each band. Moreover, with the incoherence of the beam direction, this structure shows also incoherence of the polarization. The H plan and E plan are considered here based on patch mode, which is not the case.

Response_2: Authors are grateful to the reviewer for providing the clear justification about the presented antenna. Since, it is explained in the design steps of the antenna how we have achieved the three frequency bands. In the revised manuscript we have changed the patch into monopole antenna.

Comment_3: The measured prototype: it is clear that the addition of the connector as shown in fig 5 will affect the pattern, the efficiency as well as the S11. This type of connector will add a minimum of 1.5 dB loss per the second band and 3 dB at 60 GHz. This loss plus the radiation how the measured efficiency is 85 %.

Response_3: It is right that simulations and measurements results can never be the same but they can approach each other. Since, the prototype is not fabricated only single time. To approach the optimum values of the measurement results, the prototype was fabricated and measured several times. Here, the authors want to clarify that the above-mentioned efficiency is not the measured efficiency and also not in the free space. Actually, this is the simulated radiation efficiency over the watch strap that is found to be 85%. We have updated the graphs in the revised manuscript.

Comment_4: To be considered multi-band, independent frequency control would have to be clearly demonstrated.

Response_4: Authors are grateful to the reviewer for this suggestion. We have added more demonstration in the revised manuscript.

Reviewer 2 Report

The presented results seems rather good, and with a quite close agreement between experiments and simulations. There is a small disagreement, more noticeable at higher frequencies, but is probably due to unavoidable inaccuracies, at these frequencies, in the realization and measurement (as the authors correcty noticed).

But the rest of the paper is confuse and  full of inaccuracies, and must be corrrected and rewritten.

Some of the most important flaws follow.

Abstract: the realization is done using Rogers, but it seems from the Abstract that the proposed antenna cannot be optimized with other materials. The Gain data reported here do not follow from the paper (and it is unclear here whethet are free-space or phanton ones).

Fig. 1: bottom view is missing (see row 104).

Sect. 2.2 and 2.3 must logically be inserted before Sect. 2.1. In the rpesent form, it seems a design report in which the authors have inserted some analysis and parametric sweep just to increase the appeal of the manuscript.

Sect. 3.2: What is the "40 GHZ constraint"? It seems to me that the pattern measurement setup has a limit of 40 GHz. If this is the case, please state it clearly.

row 227: in view of the "40 GHZ constraint", how is the 60 GHZ gain measured?

Effect of the numeric phantom: The proposed antenna has no ground plane, so it is impossible that the phantom gain is even larger than the free-space one. While the directive gain can (and actually, from the pattern in Fig. 10, is). What definition of efficiency have the authors used?

Fig. 8: a comparison between free-space and phantom S11 would be very useful.

Fig. 9 is missing.

Sect. 4 is completely independent from the rest of the manuscript. The cited figures show other data, the gain data are nowhere found, the efficiency when the antenna is close to the phantom cannot be almost equal to the efficiency in free-space (and is probably around 10-20%).

Table 2: Again, the gain data are different from those of sect. 3.3

Author Response

Reviewer 2

The presented results seem rather good, and with a quite close agreement between experiments and simulations. There is a small disagreement, more noticeable at higher frequencies, but is probably due to unavoidable inaccuracies, at these frequencies, in the realization and measurement (as the authors correctly noticed). But the rest of the paper is confused and full of inaccuracies, and must be corrected and rewritten. Some of the most important flaws follow.

Comment_1: Abstract: the realization is done using Rogers, but it seems from the Abstract that the proposed antenna cannot be optimized with other materials. The Gain data reported here do not follow from the paper (and it is unclear here whether are free-space or phantom ones).

Response_1: Authors are thankful to the reviewer for the point raised. It is true that the antenna is not optimized with the other material but we would like to add that we have used Roger 5880 with thickness 0.25mm upon availability of the material in the lab. The gain reported in the paper is on phantom but we forgot to add the gain vs frequency chart. But in the revised manuscript the authors have added the missing figures.

Comment_2: Fig. 1: bottom view is missing (see row 104).

Response_2: We are thankful to the reviewer for this mistake. We have added the back view in the revise manuscript.

Comment_3: Sect. 2.2 and 2.3 must logically be inserted before Sect. 2.1. In the represent form, it seems a design report in which the authors have inserted some analysis and parametric sweep just to increase the appeal of the manuscript.

Response_3: Authors are grateful for this suggestion. We have shifted the section 2.2 and section 2.3 before section 2.1.

Comment_4: Sect. 3.2: What is the "40 GHZ constraint"? It seems to me that the pattern measurement setup has a limit of 40 GHz. If this is the case, please state it clearly.

Response_4: Yes, the reviewer is correctly said that 40 GHz constraint means authors did not find the measurement setup more than 40 GHz such as radiation pattern has a limit up to 40 GHz.

Comment_5: Row 227: in view of the "40 GHZ constraint", how is the 60 GHZ gain measured?

Response_5: Authors are thankful to the reviewer for the point raised. In the view of the 40 GHz constraint, actually the gain at 60 GHz is not fully measured but we have added some points with calculations as the expected gain of the antenna at 60 GHz. But in the revised manuscript, we have updated the figure to avoid any further confusion.

Comment_6: Effect of the numeric phantom: The proposed antenna has no ground plane, so it is impossible that the phantom gain is even larger than the free-space one. While the directive gain can (and actually, from the pattern in Fig. 10, is). What definition of efficiency have the authors used?

Response_6: We are thankful to the reviewer for these comments. It is true that the CPW-fed tri-band monopole antenna does not have the ground plane on its back. But before the human body phantoms such as SKIN we have used a rubber material that sometimes also used in substrates. So, in our case, you can understand in the sense that the Rubber material working as a superstrate and thus the gain of the did not decrease. We have defined in the efficiency in the revised manuscript and highlighted it.

Comment_7:  Fig. 8: a comparison between free-space and phantom S11 would be very useful.

Response_7: We are grateful for this suggestion. We have added the comparison graph in the revised manuscript.

Comment_8: Fig. 9 is missing.

Response_8: Authors are grateful to the reviewer for pointing out the mistake. We have corrected the numbering of the Figures in the revised manuscript.

Comment_9: Sect. 4 is completely independent from the rest of the manuscript. The cited figures show other data, the gain data are nowhere found, the efficiency when the antenna is close to the phantom cannot be almost equal to the efficiency in free space (and is probably around 10-20%).

Response_9: Authors are thankful to the reviewer for this valuable suggestion. We have improved the section 4 with more explanation. Also, we have added the separate graph for the gain and efficiency when the antenna is close to the phantom.

Comment_10: Table 2: Again, the gain data are different from those of sect. 3.3.

Response_10: Authors are grateful to the reviewer for the point raised. Actually, in section 3.3 we have discussed about the gain and efficiency in free space and in the Table 2, we have added the final values when the antenna is on phantom. But in the revised manuscript, we have added separate gain graphs for the antenna on phantom to avoid any further confusion.

Reviewer 3 Report

Authors presents a discussion on CPW fed single element flexible patch antenna for millimetre wave 5G and one ISM band (60GHz). I have very important concern regarding the concept presented in this paper.

1) Are single element sufficient to meet the criteria of millimeter wave 5G band? Answer is no, then how author proposes this antenna to be deployed for millimeter wave 5G band. How you will tackle the problem of high gain, high data rate, extremely directional pattern? Until and unless you deploy array or MIMO configuration it is not possible to meet the requirment of millimeter wave 5G technology.

2) The gain and radiation pattern is big concern. You are not getting optimum directional pattern rather your single element shows dipole like behavior which certainly cannot be utilized for 5G application.

I suggest author to extend the design for array or MIMO configuration and then try to achieve the aforementioned criteria for 5G. In addition to these, following comments can be undertaken to improve single element design

1) Perform SAR analysis for the design

2) Give the effect of bending analysis

3) FBR analysis is required

4) why RT/duroid 5880 is considered? Why not other flexible substrate?

5) what is the mathematical formulation of your designed antenna? How do you obtain the first frequency? How it is related to your antenna dimension?

6) On what basis structure is chosen?

7) I want to see characteristic mode analysis (CMA) for the designed antenna.

8) How do you optimize your design?

Author Response

Reviewer 3

Authors presents a discussion on CPW fed single element flexible patch antenna for millimetre wave 5G and one ISM band (60GHz). I have very important concern regarding the concept presented in this paper.

Comment_1: Is single element sufficient to meet the criteria of millimeter wave 5G band? Answer is no, then how author proposes this antenna to be deployed for millimeter wave 5G band. How you will tackle the problem of high gain, high data rate, extremely directional pattern? Until and unless you deploy array or MIMO configuration it is not possible to meet the requirement of millimeter wave 5G technology.

Response_1: Authors are humbly thankful to the reviewer for this valuable comment. It is true that most of the single element antenna are unable to meet the criteria of the millimetre-wave 5G bands. But in our case, we have proposed triple frequency bands with single element antenna. Surely, they are not going to fully meet the requirement of 5G but at somehow, they are still acceptable as in our last case of the antenna over human body phantom, we have successfully achieved the better gain with less specific absorption rate values. In the future work, we will surely work to deploy Antenna array or MIMO configuration. During this less period of review time, authors were not able to covert the single element into antenna array or MIMO and then fabricate and measurement observations.

Comment_2: The gain and radiation pattern are big concern. You are not getting optimum directional pattern rather your single element shows dipole like behavior which certainly cannot be utilized for 5G application.

Response_2: We are thankful to the reviewer for this comment. It is true that the antenna does not have fully directional pattern but in the future work we will also try to deploy a Metasurface plane behind the antenna. Since, with this very compact size of antenna (about 4x3 mm2) and of triband, we could achieve optimum results over the phantoms.

Comment_3: I suggest author to extend the design for array or MIMO configuration and then try to achieve the aforementioned criteria for 5G.

Response_3: Authors are thankful to the reviewer for this suggestion. Since array configuration is not easy to achieve the triband but we have simulated the 2 elements MIMO configuration of the design. The results are attached below. Since, it did not show any better output.

Comment_4: In addition to these, following comments can be undertaken to improve single element design:

Comment (a): Perform SAR analysis for the design

Response_(a): Authors are thankful for this comment. Since, the current system is not compatible to calculate SAR radiation but we have used calculations to find SAR curves vs Frequency that are being added in the revised manuscript.

Comment (b): Give the effect of bending analysis

Response_(b): Authors are grateful to the reviewer for this comment. We have added the bending analysis along x-axis and y-axis of the design in the revised manuscript.

Comment (c): FBR analysis is required

Response_(c): We have added the front to back ratio (FBR) analysis in the revised manuscript.

Comment (d): why RT/duroid 5880 is considered? Why not other flexible substrate?

Response_(d): We are thankful to the reviewer for the point raised. Since, authors have used the roger 5880 substrate upon availability in the lab. This material is selected since it has a low dielectric constant and low dielectric loss, making them well suited for high frequency/broadband applications. The Rogers 5880 is isotropic and absorbs low moisture. There are many flexible materials but Roger 5880 material is one of the compatible materials that can easily be used over human body phantom.

Comment (e): what is the mathematical formulation of your designed antenna? How do you obtain the first frequency? How it is related to your antenna dimension?

Response_(e): In the revised manuscript, the authors have added the mathematical formulation of the designed antenna and explained it in detail that how we have achieved the first frequency and how it was related to the dimensions.

Comment (f): On what basis structure is chosen?

Response_(f): Authors are grateful to the reviewer for this concern.  We have already explained the design steps of the antenna in section 2.1. Since, each slot and the structure are associated to the operated frequency band. The radiating patch has two slots inside and truncated sides of the patch. The truncated sides have contribution in 28GHz, the upper invert C-shape slot helps to resonate at 38GHz while the lower U-shape slot is created to operate at 60GHz.

Comment (g): I want to see characteristic mode analysis (CMA) for the designed antenna.

Response_(g): We are thankful to the reviewer for the point raised. We have attached the results of the characteristic mode analysis (CMA) for the designed antenna.

Characteristic Angle:

Eigenvalue:

Modal Significance:

Comment (h): How do you optimize your design?

Response_(h): Authors are thankful to the reviewer for the point raised. We have optimized the design by varying various slots in the proposed antenna. In Section 2.2, we have briefly explained how we have optimized the dimensions of the various slots. We have done analysis for the dimensions such as W1, W6, W7, Lg, and Wg. After optimizing these slots, it is clear that the proposed tri band can be fully adjusted by varying the values of the abovementioned parameters.

Round 2

Reviewer 1 Report

The authors have made a good effort to improve the quality.

The control of the frequency/bandwidth mechanism is unclear. How I can shift frequency 2 by setting frequencies 1 and 3?

the bonding along the Y/X_axis should not affect the matrix. But it wins. Please check and comment.

Author Response

The authors have made a good effort to improve the quality.

Comment_1: The control of the frequency/bandwidth mechanism is unclear. How can I shift frequency 2 by setting frequencies 1 and 3?

Response_1: We are thankful to the reviewer for this suggestion and for your valuable time. In section 2.1 and 2.2, authors have explained the design procedure and the parametric analysis, in which we have clearly mentioned that the antenna has three main slots; one is truncated sides (for 28 GHz), second is upper C-shaped slot (for 38 GHz), and the third is lower U-shaped slot (for 60 GHz). In the parametric study, we have analyzed and explained that by varying the lengths and widths of the above slots we can shift the frequencies. So, each slot in the radiator is independent to the other. We can easily shift one frequency by setting the rest of the frequencies. We have further elaborated this comment in the revised manuscript (sect. 2.2)

Comment_2: The bending along the Y/X_axis should not affect the matrix. But it wins. Please check and comment.

Response_2: We are thankful to the reviewer for this valuable comment. In section 2.5, we have explained the bending analysis of the antenna along y-axis and x-axis. Here, we want to clarify that the size of the proposed antenna is very compact (4 x 3 x 0.25mm3). With this compact size, the bending radii seem to be negligible even if we place the antenna on phantoms. In the manuscript, we just have shown the pictures (Fig. 5a & 6a) clearer to show to the reviewer and to the readers (for the ease of understand) that how the antenna looks like if it is in bent condition.

Reviewer 2 Report

Some flaws of version 1 have been reasonably corrected. But there are still several errors.

Main errors follows.

Abstract: RT/Duroid comes "out of the blue" here. It is clear that it is the material used for experimental tests. And so, it is correct that it has been used for the optimization procedure. But it must be introduced there (around row 140 or, better, row 185 where it is actuallly used) and not in the Abstract.

row 90: printed monopole antennas are not considered in [21]. So, a different reference must be included.
And, moreover, when a book is used as a reference, it is very useful to include also the referenced section (or the pages).

row 120-144: all these equations are about a printed patch witha ground plane, and therefore are completely wrong for a printed monopole without ground plane. And infact, using (2-4) one gets L=3.74 mm (not 2.4 mm as claimed by the authors).

row 123: ref. [31-32] are absent.

Sect. 3.3: the definition of efficiency is repeated three times.

Fig. 9: from Fig. 8 it is clear that the FBR is around 0 dB. The data in Fig. 9b are therefore wrong.

Sect. 4: According to the Gabriel's data, reported in  "Hall P.S., Hao Y. - Antennas And Propagation for Body-Centric Wireless Communications-Artech House  2nd ed (2012)  ", sect. 2.1, the conductifity of the dry skin is around 30 S/m (not 0.5 S/m) and the real part is significantly smaller than those reported in the paper (around 15 to 20).  Therefore, the presented simulations are completely useless to evaluate the antenna behavior as a smart watch antenna. 

Author Response

Reviewer 2

Some flaws of version 1 have been reasonably corrected. But there are still several errors.

Main errors follows.

Comment_1: Abstract: RT/Duroid comes "out of the blue" here. It is clear that it is the material used for experimental tests. And so, it is correct that it has been used for the optimization procedure. But it must be introduced there (around row 140 or, better, row 185 where it is actually used) and not in the Abstract.

Response_1: Authors appreciate the reviewer suggestions. We have shifted the RT/Duroid sentence to the row 185 where it is actually used. The comment has been addressed in the revised manuscript.

Comment_2: row 90: printed monopole antennas are not considered in [21]. So, a different reference must be included. And, moreover, when a book is used as a reference, it is very useful to include also the referenced section (or the pages).

Response_2: We are thankful to the reviewer for pointing out the mistake. We have replaced a new reference on Printed Monopole in the revised manuscript.

Comment_3: row 120-144: all these equations are about a printed patch with a ground plane, and therefore are completely wrong for a printed monopole without ground plane. And in fact, using (2-4) one gets L=3.74 mm (not 2.4 mm as claimed by the authors).

Response_3: We are thankful to the reviewer for this valuable comment. We have removed all the equations and put the correct equations used for the CPW-fed printed monopole antenna in the revised manuscript.

Comment_4: row 123: ref. [31-32] are absent.

Response_4: Authors have mistakenly added ref [31-32]. We have removed this in the revised manuscript.

Comment_5: Sect. 3.3: the definition of efficiency is repeated three times.

Response_5: Authors have removed the repeated lines in the revised manuscript.

Comment_6: Fig. 9: from Fig. 8 it is clear that the FBR is around 0 dB. The data in Fig. 9b are therefore wrong.

Response_6: We are thankful to the reviewer for this worthy comment. Actually in Fig.8, the authors have normalized the radiation pattern to 0dB. While in Fig.9, it is the simulated FBR graph that we obtained after simulation of the design.

Comment_7: Sect. 4: According to the Gabriel's data, reported in "Hall P.S., Hao Y. - Antennas And Propagation for Body-Centric Wireless Communications-Artech House 2nd ed (2012)  ", sect. 2.1, the conductivity of the dry skin is around 30 S/m (not 0.5 S/m) and the real part is significantly smaller than those reported in the paper (around 15 to 20).  Therefore, the presented simulations are completely useless to evaluate the antenna behavior as a smart watch antenna. 

Response_7: Authors appreciate the reviewer’s point raised. Actually, authors have used the rubber layer behind the antenna that do not allow to effect on the resonance of the antenna. The conductivity of the skin layer varies from different body parts. Also, the antenna is designed at mm-wave band at which the conductivity of the skin may also vary. The antenna is being simulated on the smart watch strap in order to know that whether the antenna is stable on the watch strap or not if it is practically implemented in our future works. Since, the antenna has shown optimum and stable results even on the watch strap therefore authors claimed that the antenna is suitable for smart watch applications at mm-wave band.

Reviewer 3 Report

Thanks for replying to my comments. As the author is unable to incorporate my comments due to limited time and facility, I do not have any further comments.

Author Response

Authors are grateful to the reviewer for your valuable time.